# On component interactions in two-stage recommender systems

**Jiri Hron**
University of Cambridge

**Karl Krauth**
UC Berkeley

**Michael I. Jordan**
UC Berkeley

**Niki Kilbertus**
Technical University of Munich
Helmholtz AI, Munich

## Abstract

Thanks to their scalability, two-stage recommenders are used by many of today's largest online platforms, including YouTube, LinkedIn, and Pinterest. These systems produce recommendations in two steps: (i) multiple *nominators*—tuned for low prediction latency—preselect a small subset of candidates from the whole item pool; (ii) a slower but more accurate *ranker* further narrows down the nominated items, and serves to the user. Despite their popularity, the literature on two-stage recommenders is relatively scarce, and the algorithms are often treated as mere sums of their parts. Such treatment presupposes that the two-stage performance is explained by the behavior of the individual components in isolation. This is not the case: using synthetic and real-world data, we demonstrate that interactions between the ranker and the nominators substantially affect the overall performance. Motivated by these findings, we derive a generalization lower bound which shows that independent nominator training can lead to performance on par with uniformly random recommendations. We find that careful design of item pools, each assigned to a different nominator, alleviates these issues. As manual search for a good pool allocation is difficult, we propose to learn one instead using a Mixture-of-Experts based approach. This significantly improves both precision and recall at $K$.

## 1  Introduction

Recommender systems play a central role in online ecosystems, affecting what media we consume, which products we buy, or even with whom we interact. A key technical challenge is ensuring these systems can sift through billions of items to deliver a personalized experience to millions of users with *low response latency*. A widely adopted solution to this problem are two-stage recommender systems [12, 20, 28, 98, 101] where (i) a set of computationally efficient *nominators* (or *candidate generators*) preselects a small number of candidates, which are then (ii) further narrowed down, reranked, and served to the user by a slower but more statistically accurate *ranker*.

Nominators are often heterogeneous, ranging from associative rules to recurrent neural networks [18]. A popular choice are matrix factorization [56, 72] and two-tower [98] architectures which model user feedback by the dot product between user and item embeddings. While user embeddings often evolve with the changing context of user interactions, item embeddings can typically be precomputed before deployment. The cost of candidate generation is thus dominated by the (approximate) computation of the embedding dot products. In contrast, the ranker often takes *both* the user and item features as input, making the computational cost linear in the number of items even at deployment [20, 67].

35th Conference on Neural Information Processing Systems (NeurIPS 2021).

With few exceptions [44, 52, 67], two-stage specific literature is sparse compared to that on *single-stage* systems (i.e., recommenders which do not construct an explicit candidate set within a separate nominator stage [e.g., 17, 40, 46, 56, 72, 78, 84]). This is especially concerning given the considerable ethical challenges entailed by the enormous reach of two-stage systems: according to the recent systematic survey by Milano et al. [70], recommender systems have been (partially) responsible for unfair treatment of disadvantaged groups, privacy leaks, political polarization, spread of misinformation, and 'filter bubble' or 'echo chamber' effects. While many of these issues are primarily within the realm of 'human–algorithm' interactions, the additional layer of 'algorithm–algorithm' interactions introduced by the two-stage systems poses a further challenge to understanding and alleviating them.

The main aim of our work is thus to narrow the knowledge gap between single- and two-stage systems, particularly in the context of *score-based* algorithms. Our main contributions are:

1. We show two-stage recommenders are significantly affected by interactions between the ranker and the nominators over a variety of experimental settings (Section 3.1).

2. We investigate these interactions theoretically (Section 3.2), and find that while independent ranker training typically works well (Proposition 1), the same is not the case for the nominators where two popular training schemes can both result in performance no better than that of a uniformly random recommender (Proposition 2).

3. Responding to the highlighted issues with *independent* training, we identify specialization of nominators to smaller subsets of the item pool as a source of potentially large performance gains. We thus propose a *joint* Mixture-of-Experts [47, 49] style training which treats each nominator as the expert for its own item subset. The ability to learn the item pool division alleviates the issues caused by the typically lower modeling capacity of the nominators, and empirically leads to improved precision and recall at K (Section 4).

## 2 Two-stage recommender systems

The goal of recommender systems is to learn a policy $\pi$ which maps contexts $x \in \mathcal{X}$ to distributions $\pi(x)$ over a finite set of items (or *actions*) $a \in \mathcal{A}$, such that the expected reward $\mathbb{E}_x \mathbb{E}_{a \sim \pi(x)}[r_a \mid x]$ is maximized. The context $x$ represents information about the user and items (e.g., interaction history, demographic data), and $r_a$ is the user feedback associated with item $a$ (e.g., rating, clickthrough, watch-time). We assume that $r_a|x$ has a well-defined fixed mean $f^\star(x, a) := \mathbb{E}[r_a \mid x]$ for all the $(x, a)$ pairs. To simplify, we further assume only one item $a_t$ is to be recommended for each given context $x_t$, where $t \in [T]$ with $[T] := \{1, \ldots, T\}$ for $T \in \mathbb{N}$.

Two-stage systems differ from the single-stage ones by the two-step strategy of selecting $a_t$. First, each nominator $n \in [N]$ picks a single candidate $a_{n,t}$ from its *assigned pool* $\mathcal{A}_n \subseteq \mathcal{A}$ ($\mathcal{A}_n \neq \varnothing$, $\bigcup_n \mathcal{A}_n = \mathcal{A}$). Second, the ranker chooses an item $a_t$ from the *candidate pool* $\mathcal{C}_t := \{a_{1,t}, \ldots, a_{N,t}\}$, and observes the reward $r_t = r_{ta_t}$ associated with $a_t$. Since the goal is *expected* reward maximization, recommendation quality can be measured by *instantaneous regret* $r_t^\star - r_t$ where $r_t^\star = r_{ta_t^\star}$ is the reward associated with an optimal arm $a_t^\star \in \operatorname{argmax}_{a \in \mathcal{A}} f^\star(x_t, a)$. This leads us to an important identity for the *(cumulative) regret* in two-stage systems which is going to be used throughout:

$$\mathrm{R}_T^{2\mathtt{s}} = \sum_{t=1}^T r_t^\star - r_t = \underbrace{\sum_{t=1}^T (r_t^\star - r_{t\tilde{a}_t})}_{=:\mathrm{R}_T^{\mathtt{N}}} + \underbrace{\sum_{t=1}^T (r_{t\tilde{a}_t} - r_t)}_{=:\mathrm{R}_T^{\mathtt{R}}}, \tag{1}$$

with $\tilde{a}_t \in \operatorname{argmax}_{a \in \mathcal{C}_t} f^\star(x_t, a)$. In words, $\mathrm{R}_T^{\mathtt{N}}$ is the *nominator regret*, quantifying the difference between the best action presented to the ranker $\tilde{a}_t$ and the best overall action $a_t^\star$, and $\mathrm{R}_T^{\mathtt{R}}$ is the *ranker regret* which measures the gap between the choice of the ranker $a_t$ and the best action in the candidate set $\mathcal{C}_t$. The two-stage recommendation process is summarized in Figure 2 (left).

While Equation (1) is typical for the *bandit* literature (data collected interactively, only $r_{ta_t}$ revealed for each $t$), we also consider the *supervised* learning case (a dataset with all $\{r_{ta}\}_{ta}$ revealed is given). In particular, Section 3 presents an empirical comparison of single- and two-stage systems in the bandit setting, followed by a theoretical analysis with implications for both the learning setups.

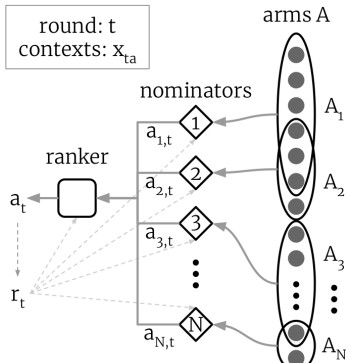
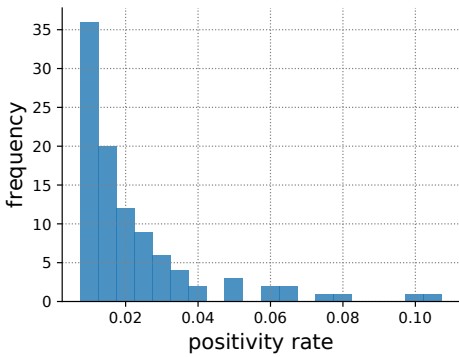

Figure 2: **Left:** The two-stage recommendation setup. **Right:** Amazon reward histogram. The top 5 arms are responsible for 19.22% whereas the bottom 50 only for 19.85% of the positive rewards.

## 3 Comparing single- and two-stage systems

Since the ranker and nominators could each be deployed independently, one may wonder whether the performance of a two-stage system is significantly affected by factors beyond the hypothetical single-stage performance of its components. This question is both theoretically (new developments needed?) and practically interesting (e.g., training components independently, as common, assumes targeting single-stage behavior is optimal). In Section 3.1, we empirically show that while factors known from the single-stage literature also affect two-stage systems, there are *two-stage specific properties* which can be even more important. Section 3.2 then investigates these properties theoretically, revealing a non-trivial interaction between the nominator training objective and the item pool allocations $\{\mathcal{A}_n\}_n$.

### 3.1 Empirical observations

**Setup**

We study the effects of item pool size, dimensionality, misspecification, nominator count, and the choice of ranker and nominator algorithms *in the bandit setting*. We compare single- and two-stage systems where each (component) models the expected reward as a *linear* function $f_t(x, a) = \langle \hat{\theta}_t, x_a \rangle$ ($x_a$ differs for each $a$). Abbreviating $x_t = x_{ta_t}$, the estimates are converted into a policy via either:

1. **UCB (U)** [6, 64] which computes the ridge-regression estimate with regularizer $\lambda > 0$

$$\hat{\theta}_t := \Sigma_t \sum_{i=1}^{t-1} x_i r_i, \qquad \Sigma_t := \left(\lambda I + \sum_{i=1}^{t-1} x_i x_i^\top\right)^{-1},  \tag{2}$$

and selects actions with exploration bonus $\alpha > 0$: $a_t \in \mathrm{argmax}_a \langle \hat{\theta}_t, x_{ta} \rangle + \alpha \sqrt{x_{ta}^\top \Sigma_t x_{ta}}$.

2. **Greedy (G)** [8, 53] which can be viewed as a special case of UCB with $\alpha = 0$.

The $\mathrm{argmax}$ is restricted to $\mathcal{A}_n$ (resp. $\mathcal{C}_t$) in two-stage systems, with pool allocation $\{\mathcal{A}_n\}_n$ designed to minimize overlaps and approximately equalize the number of items in each pool (see Appendix B).

We chose to make the above restrictions of our experimental setup to limit the large number of design choices two-stage recommenders entail (architecture and hyperparameters of each nominator and the ranker, item pool allocation, number of nominators, etc.), and with that isolate the variation in performance to only a few factors of immediate interest.

We use one synthetic and one real-world dataset. The **synthetic dataset** is generated using a linear model $r_{ta} = \langle \theta^\star, x_{ta} \rangle + \varepsilon_{ta}$ for each time step $t \in [T]$ and action $a \in \mathcal{A}$. The vector $\theta^\star$ is drawn uniformly from the $d$-dimensional unit sphere at the beginning and then kept fixed, the contexts $x_{ta}$ are sampled independently from $\mathcal{N}(0, I)$, and $\varepsilon_{ta} \sim \mathcal{N}(0, 0.01)$ is independent observation noise.

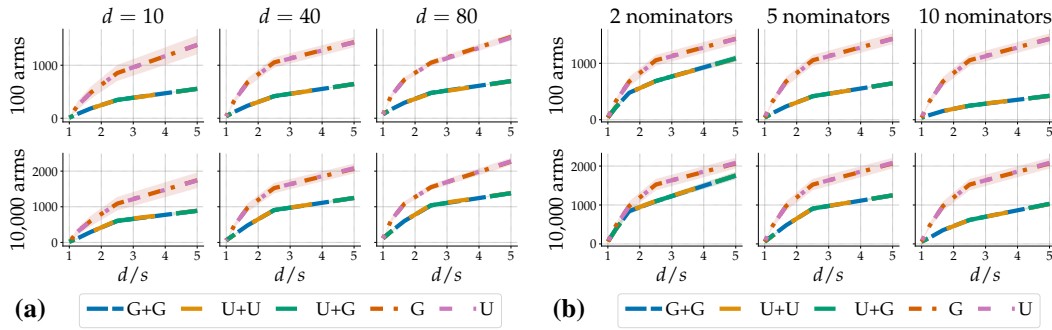

Figure 3: Synthetic data results. The x-axis is the ratio between the true feature dimension $d$ and the size of the subset available to the nominators and the single-stage systems $s$. The y-axis shows the expected regret at $T = 1000$. In plot **(a)**, $N = 5$ nominators are used, and columns represent the total number of features $d$. In plot **(b)**, $d = 40$ features are used, and columns show the number of nominators $N$. The legend describes model architectures, where two-stage systems are labeled by '`[ranker]`+`[nominator]`' (e.g., '`U+G`' is a UCB ranker with Greedy nominators).

The **real-world dataset** 'AmazonCat–13K' contains Amazon reviews and the associated product category labels [10, 69].[1] Since 'AmazonCat–13K' is a multi-label classification dataset, we convert it into a bandit one by assigning a reward of one for correctly predicting any one of the categories to which the product belongs, and zero otherwise. An $|\mathcal{A}|$–armed linear bandit is then created by sampling $|\mathcal{A}|$ reviews uniformly from the whole dataset, and treating the associated features as the contexts $\{x_{ta}\}_{a \in \mathcal{A}}$. This method of conversion is standard in the literature [27, 35, 66, 67].

We use only the raw text features, and convert them to 768-dimensional embeddings using the HuggingFace pretrained model '`bert-base-uncased`' [24, 97];[2] we further subset to the first $d = 400$ dimensions of the embeddings, which does not substantially affect the results. Because we are running thousands of different experiment configurations (counting the varying seeds), we further reduce the computational complexity by subsetting from the overall 13K to only 100 categories. Since most of the products belong to 1–3 categories, we take the categories with 3[rd] to 102[nd] highest occurrence frequency. This ensures less than 5% of the data points belong to none of the preserved categories, and overall 10.73% reward positivity rate with strong power law decay (Figure 2, right).

While the ranker can always access all $d$ features, the usual lower flexibility of the nominators (*misspecification*) is modelled by restricting each to a different *random subset* of $s$ out of the total $d$ features on both datasets. This is equivalent to forcing the corresponding regression parameters to be zero. Both UCB and Greedy are then run with $x_t$ replaced by the $s$-dimensional $x_{n,t} = x_{n,ta_t}$ everywhere. The same restriction is applied to the single-stage systems for comparison. In all rounds, each nominator is updated with $(x_{n,t}, a_t, r_t)$, regardless of whether $a_t \in \mathcal{A}_n$ (this assumption is revisited in Section 3.2). Thirty independent random seeds were used to produce the (often barely visible) two-sigma standard error regions in Figures 3 and 4. More details—including the hyperparameter tuning protocol and additional results—can be found in Appendices B and C.

### Results

Starting with the synthetic results in Figure 3, we see that the number of arms and the feature dimension $d$ are both correlated with increased regret in single- *and* two-stage systems. Another similarity between all the algorithms is that misspecification—as measured by $d/s$—also has a significant effect on performance.[3] This is also the case for the Amazon dataset in Figure 4.

---

[1]We did not use 'MovieLens' [38] since it contains little useful contextual information as evidenced by its absence even from the state-of-the-art models [79]. Two-stage recommenders are only used when context matters as otherwise all recommendations could be precomputed and retrieved from a database at deployment.

[2]Encoded dataset: `https://twostage.s3-us-west-2.amazonaws.com/amazoncat-13k-bert.zip`.

[3]Misspecification error typically translates into a linear regret term $\epsilon T$ [21, 25, 31, 32, 36, 58, 61]. We can thus gain some *intuition* for the concavity of $d/s \mapsto R_T$ from the $L^2$ error $\epsilon = \min_{\theta_n} (\mathbb{E}[(r_a - \langle \theta_n, x_{n,a} \rangle)^2])^{1/2}$ where $a \sim \text{Unif}(\mathcal{A})$ [58]. Using $x_{ta} \sim \mathcal{N}(0, I)$, the minimum is achieved by $(\mathbb{E}[x_{n,a} x_{n,a}^\top])^{-1} \mathbb{E}[x_{n,a} r_a] = \theta_n^\star$,

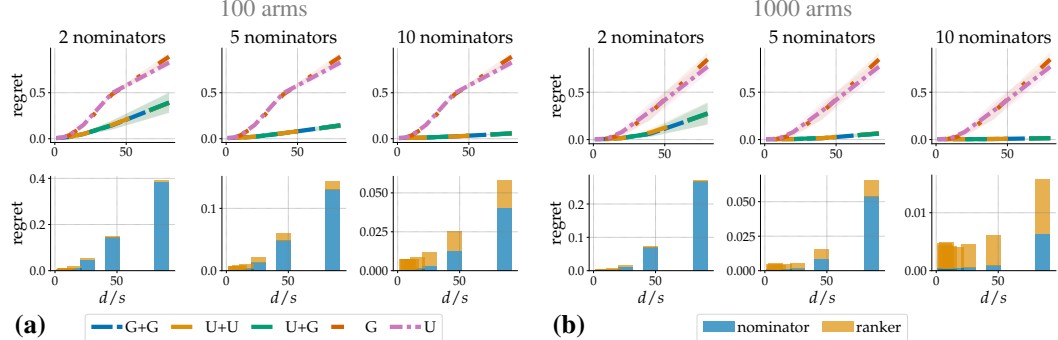

Figure 4: Amazon data results. The axes are the same as in Figure 3, except the $y$-axis is plotted at $T = 5000$ with the regret divided by that of a uniformly random agent. The feature dimension is fixed to $d = 400$, and the number of arms to 100 in plot (a), and to 1000 in plot (b). The columns represent varying number of nominators $N$. The legend is shared, where the one in (a) corresponds to the top row plots and has the same interpretation as in Figure 3, and the one in (b) belongs to the bottom row plots which show the proportion of ranker and nominator regret (see Equation (1)) for the 'U+U' two-stage system as a representative example (all two-stage systems perform similarly here).

The influence of the number of arms, dimensionality, and misspecification on single-stage systems is well known [60]. Figures 3 and 4 suggest similar effects also exist for two-stage systems. On the other hand, while the directions of change in regret agree, the magnitudes do not. In particular, two-stage systems perform significantly better than their single-stage counterparts. This is possible because the ranker can exploit its access to all $d$ features to improve upon even the best of the nominators (recall that nominators and single-stage systems share the same model architecture). In other words, *the single-stage performance of individual components does not fully explain the two-stage behavior*.

To develop further intuition about the differences between single- and two-stage systems, we turn our attention to the Amazon experiments in Figure 4. The top row suggests the performance of two-stage systems improves as the number of nominators grows. Strikingly, the accompanying UCB ranker + nominator plots in the bottom row show the nominator regret $\mathrm{R}_T^{\mathtt{N}}$ dominates when there are few nominators, but gives way to the ranker regret $\mathrm{R}_T^{\mathtt{R}}$ as their number increases.

To explain why, first note that the single-stage performance of the ranker can be read off from the bottom left corner of each plot where $d = s$ (because all the components are identical at initialization, and then updated with the same data). Since the size of the candidate set $\mathcal{C}_t$ increases with the number of nominators, the two-stage performance in the $d > s$ case eventually approaches that of the single-stage UCB ranker as well, even if the nominators are no better than random guessing. In fact, because $\approx 10\%$ of the items yield optimal reward, the probability that a set of ten uniformly random nominators with non-overlapping item pools nominates at least one optimal arm is on average $1 - \left(\frac{9}{10}\right)^{10} \approx 0.65$, i.e., the instantaneous nominator regret would be zero 65% of the time.

To summarize, we have seen evidence that properties known to affect single-stage performance—number of arms, feature dimensionality, misspecification—have similar qualitative effects on two-stage systems. However, two-stage recommenders perform significantly better than any of the nominators alone, especially as the nominator count and the size of the candidate pool increase. Complementary to the evidence from offline learning [67], and the effect of ranker pretraining [44], these observations add to the case that two-stage systems should not be treated as just the sum of their parts. We add theoretical support to this argument in the next section.

### 3.2 Theoretical observations

The focus of Section 3.1 was on linear models in the bandit setting. We lift the bandit assumption later in this section, and relax the class of studied models to *least-squares regression oracle* (LSO) based algorithms, which estimate the expected reward by minimizing the sum of squared errors and a

---

with $\theta_n^\star$ the $s$ entries of $\theta^\star$ corresponding to the dimensions available to $n$. The $L^2$ error is thus a concave function of $d/s$ by symmetry: $\epsilon = \sqrt{\mathbb{E}[\mathbb{E}[(r_{ta} - \langle \theta_n^\star, x_{n,ta} \rangle)^2 \mid \theta^\star]]} = \sqrt{\mathbb{E}[\|\theta^\star\|_2^2 - \|\theta_n^\star\|_2^2]} = \sqrt{1 - s/d}$.

regularizer $\| \cdot \|_{\mathcal{F}}$ over a given model class $\mathcal{F}$

$$f_t \in \operatorname{argmin}_{f \in \mathcal{F}} \left\{ \|f\|_{\mathcal{F}} + \sum_{i=1}^{t-1} (r_i - f(x_i, a_i))^2 \right\}. \tag{3}$$

These estimates are then converted into a policy either greedily, $\pi_t(x) = \operatorname{Unif}(\operatorname{argmax}_a f_t(x, a))$, or by incorporating an exploration bonus as in LinUCB [6, 19, 22, 64, 82], or the more recent class of black-box reductions from bandit to online or offline regression [30–32, 58, 86]. The resulting algorithms are often minimax optimal, and (some) also perform well on real-world data [11].

We choose LSO based algorithms because they (i) include the Greedy and (Lin)UCB models studied in the previous section, and (ii) allow for an easier exposition than the similarly popular cost-sensitive classification approaches [e.g., 2, 18, 20, 26, 59]. The following proposition is an application of the fact that algorithms like LinUCB or SquareCB [1, 30, 32] provide regret guarantees robust to contexts chosen by an *adaptive* adversary, and thus also to those chosen by the nominators.

**Proposition 1** *Assume the ranker achieves a* single-stage *regret guarantee* $R_T \leq B_T^{\text{R}}$ *for some constant* $B_T^{\text{R}} \in \mathbb{R}$ *(either in expectation or with high probability), even if the contexts* $\{x_t\}_{t=1}^{T}$ *are chosen by an adaptive adversary. The ranker regret then satisfies*

$$R_T^{\text{R}} = \sum_{t=1}^{T} r_{t\tilde{a}_t} - r_t \leq B_T^{\text{R}},$$

*in the sense of the original bound (i.e., in expectation, or with high probability).*

While proving Proposition 1 is straightforward, its consequences are not. First, if $B_T^{\text{R}}$ is in some sense optimal, then Equation (1) implies the two-stage regret $R_T^{\text{2s}}$ will be dominated by the nominator regret $R_T^{\text{N}}$ (unless it satisfies a similar guarantee). Second, $R_T^{\text{R}} \leq B_T^{\text{R}}$ holds exactly when the ranker is trained in the 'single-stage mode', i.e., the tuples $(x_t, a_t, r_t)$ are fed to the algorithm without any adjustment for the fact $\mathcal{C}_t$ is selected by a set of adaptive nominators from the whole item pool $\mathcal{A}$.

The above however does not mean that the ranker has no substantial effect on the overall behavior of the two-stage system. In particular, the feedback observed by the ranker also becomes the feedback observed by the nominators, which has the *primary effect* of influencing the nominator regret $R_T^{\text{N}}$, and the *secondary effect* of influencing the candidate pools $\mathcal{C}_t$ (which creates a feedback loop). The rest of this section focuses on the primary effect, and in particular its dependence on how the nominators are trained and the item pools $\{\mathcal{A}_n\}_n$ allocated.

**Pitfalls in designing the nominator training objective**

The *primary effect* above stems from a key property of two-stage systems: unlike the ranker, nominators do not observe feedback for all items they choose. While importance weighting can be used to adjust the nominator training objective [67], it does not tell us what adjustment would be optimal.

We thus study two major types of updating strategies: (i) 'training-on-all,' and (ii) 'training-on-own.' Both can be characterized in terms of the following weighted oracle objective for the $n^{\text{th}}$ nominator

$$f_{n,t} \in \operatorname{argmin}_{f_n \in \mathcal{F}_n} \left\{ \|f_n\|_{\mathcal{F}_n} + \sum_{i=1}^{t-1} w_{n,i}(r_i - f_n(x_{n,i}, a_i))^2 \right\}, \tag{4}$$

where $\mathcal{F}_n$ is the class of functions the nominator can fit, $\| \cdot \|_{\mathcal{F}_n}$ the regularizer, and $w_{n,t} = w_{n,a_t} \geq 0$ the weight. **'Training-on-all'**—used in Section 3.1—takes $w_{n,a} = 1$ for all $(n, a)$, which means *all* data points are valued equally regardless of whether a particular $a_t$ belongs to the nominator's pool $\mathcal{A}_n$. 'Training-on-all' may potentially waste the already limited modelling capacity of the nominators if the pools $\mathcal{A}_n$ are not identical. The **'training-on-own'** alternative therefore uses $w_{n,a} = \mathbb{1}\{a \in \mathcal{A}_n\}$ so that only the data points for which $a_t \in \mathcal{A}_n$ influence the objective.[4]

While 'training-on-all' and 'training-on-own' are not the only options we could consider, they are representative of two very common strategies. In particular, 'training-on-all' is the default easy-to-implement option which sometimes performs surprisingly well [11, 81]. In contrast, 'training-on-own'

---

[4]There are two possible definitions of 'training-on-own': (i) $w_{n,t} = \mathbb{1}\{a_t \in \mathcal{A}_n\}$; (ii) $w_{n,t} = \mathbb{1}\{a_t = a_{n,t}\}$. While the main text considers the former, Proposition 2 can be extended to the latter with minor modifications.

approximates the (on-policy) 'single-stage mode' where the nominator observes feedback only for the items it selects (in particular, $a_t \in \mathcal{A}_n$ only if $a_t = a_{n,t}$ when the pools are non-overlapping).

Proposition 2 below shows that neither 'training-on-all' nor 'training-on-all' is guaranteed to perform better than random guessing in the infinite data limit ($T \to \infty$). We consider the *linear* setting $f^\star(x_t, a) = \langle \theta^\star, x_{ta} \rangle$ for all $(x_t, a)$, $\theta^\star$ fixed, with nominators using ridge regression oracles $f_{n,t}(x_{n,t}, a) = \langle \hat{\theta}_{n,t}, x_{n,ta} \rangle$ as defined in Equation (2), $\lambda \geq 0$ fixed, and $x_{n,ta}$ again a *subset* of the full feature vector $x_{ta}$. We also assume the nominators take the predicted best action with non-vanishing probability (Assumption 1), which holds for all the cited LSO based algorithms.

**Assumption 1** *Let $f_{n,t}$ be as in Equation (4), and denote $\mathcal{A}_{n,t}^{\mathsf{G}} := \mathrm{argmax}_{a \in \mathcal{A}_n} f_{n,t}(x_{n,t}, a)$. We assume there is a universal constant $\delta > 0$ such that for all $n \in [N]$ and $a \in \mathcal{A}_n$ with $\limsup T^{-1} \sum_{1 \leq t \leq T} \mathbb{P}(a_t^\star = a) > 0$, we have $\limsup T^{-1} \sum_{1 \leq 1 \leq T} \mathbb{P}(a_{n,t} \in \mathcal{A}_{n,t}^{\mathsf{G}} \mid a_t^\star = a) \geq \delta$.*

**Proposition 2** *In both the supervised and the bandit learning setup, there exist two* distinct *context distributions with pool allocations $\{\mathcal{A}_n\}_n$, and $r_a \in [0, 1]$ almost surely (a.s.) for all $a \in \mathcal{A}$, such that 'training-on-own' (resp. 'training-on-all') leads to asymptotically linear two-stage regret*

$$\limsup_{T \to \infty} \frac{\mathbb{E}[\mathrm{R}_T^{2\mathrm{s}}]}{T} > 0 \,.$$

*Moreover, the asymptotic regret of 'training-on-all' is sublinear under the context distribution and pool allocation where 'training-on-own' suffers linear regret, and vice versa.*

**Proof 1** *Throughout, we use $\hat{\theta}_{n,t} \to (\mathbb{E}[x_{n,a} x_{n,a}^\top])^{-1} \mathbb{E}[x_{n,a} r_a] =: \theta_n^\star$ (a.s.) by Lemma 1 (Appendix A), assuming invertibility and that $a_t$ is i.i.d.; note $\theta_n^\star = \mathrm{argmin}_{\theta_n} \mathbb{E}[(r_a - \langle \theta_n, x_{n,a} \rangle)^2]$. We allow any zero mean reward noise which satisfies $r_a \in [0, 1]$ (a.s.) for all $a \in \mathcal{A}$.*

*(I) Supervised setup. Take two nominators, $\mathcal{A}_1 = \{a_{(1)}\}$, $\mathcal{A}_2 = \{a_{(2)}, a_{(3)}\}$, a single context*

$$X := \begin{bmatrix} \text{---} & x_{a_{(1)}} & \text{---} \\ \text{---} & x_{a_{(2)}} & \text{---} \\ \text{---} & x_{a_{(3)}} & \text{---} \end{bmatrix} = \begin{bmatrix} 1 & 0 & -1 \\ 0 & 1 & 0 \\ 0 & 0 & 1 \end{bmatrix}, \tag{5}$$

*and restrict the nominators to the last two columns of $X$. As $|\mathcal{A}_1| = 1$, the first nominator always proposes $a_{(1)}$, disregards of its fitted model. Since all rewards are revealed and used to update the model in the supervised setting, 'training-on-all' $t \to \infty$ limit for the second nominator's $\hat{\theta}_{2,t}$ is*

$$\theta_2^\star = \mathrm{argmin}_{\beta \in \mathbb{R}^2} \mathbb{E}_{a \sim Unif(\mathcal{A})}[(r_a - \langle \beta, x_{2,a} \rangle)^2]$$
$$= \mathrm{argmin}_{\beta \in \mathbb{R}^2} \{(\bar{r}_1 + \beta_2)^2 + (\bar{r}_2 - \beta_1)^2 + (\bar{r}_3 - \beta_2)^2\} = [\bar{r}_2, \tfrac{\bar{r}_3 - \bar{r}_1}{2}]^\top,$$

*where $\bar{r}$ is the mean reward vector for the single context ($\theta^\star$ is then $X^{-1}\bar{r}$). If we take, e.g., $\bar{r} = [\tfrac{1}{4}, \tfrac{1}{2}, 1]^\top$, then $a_{(3)} \neq \mathrm{argmax}_{a \in \mathcal{A}_2} \langle \theta_2^\star, x_{2,a} \rangle = a_{(2)}$. On the other hand, 'training-on-own' would yield $\theta_2^\star = [\bar{r}_2, \bar{r}_3]^\top$, and thus correctly identify $a_{(3)}$ via $\mathrm{argmax}_{a \in \mathcal{A}_2} \langle \theta_2^\star, x_{2,a} \rangle$.*

*In contrast, consider the modified setup $\mathcal{A}_1 = \{a_{(1)}\}$, $\mathcal{A}_2 = \{a_{(2)}, a_{(3)}, a_{(4)}\}$*

$$X := \begin{bmatrix} \text{---} & x_{a_{(1)}} & \text{---} \\ \text{---} & x_{a_{(2)}} & \text{---} \\ \text{---} & x_{a_{(3)}} & \text{---} \\ \text{---} & x_{a_{(4)}} & \text{---} \end{bmatrix} = \begin{bmatrix} 1 & 0 & 0 & -1 \\ 0 & 1 & 0 & -1 \\ 0 & 0 & 1 & 0 \\ 0 & 0 & 0 & 1 \end{bmatrix}. \tag{6}$$

*Restricting nominators to the last two columns of $X$, 'training-on-own' would yield $\theta_2^\star = [\bar{r}_3, \tfrac{\bar{r}_4 - \bar{r}_2}{2}]^\top$ under full feedback access, whereas 'training-on-all' would converge to $\theta_2^\star = [\bar{r}_3, \tfrac{\bar{r}_4 - \bar{r}_2 - \bar{r}_1}{3}]^\top$. Hence with, e.g., $\bar{r} = [\tfrac{3}{4}, 1, \tfrac{1}{6}, \tfrac{7}{8}]^\top$, 'training-on-own' would make the second nominator pick $a_{(3)}$ via argmax, but 'training-on-all' would successfully identify the optimal $a_{(2)}$.*

*(II) Bandit setup. Take $X$ from Equation (6), but use $\bar{r} = [\tfrac{3}{4}, \tfrac{7}{8}, \tfrac{1}{6}, 1]^\top$ and the associated $\theta^\star = X^{-1}\bar{r}$. For each $j \in [4]$, let $X_{(j)}$ be a deterministic context matrix which is the same as $X$ except all but the $j^{th}$ row are replaced by zeros. Observe that for each $j$, the mean reward vector $X_{(j)}\theta^\star$ has exactly one strictly positive component, and thus $a_t^\star = a_{(j)}$ when $X_{(j)}$ is drawn.*

*Let Unif($\{X_{(j)}\}_j$) be the context distribution, $\mathcal{A}_1 = \{a_{(1)}\}$, $\mathcal{A}_2 = \{a_{(2)}, a_{(3)}, a_{(4)}\}$, and restrict nominators to the last two columns of each sampled $X_{(j)}$. We employ a proof by contradiction. Assume $\limsup T^{-1} \mathbb{E}[\mathrm{R}_T^{2s}] \to 0$. Then $\hat{\theta}_{2,t} \to \theta_2^\star$ in probability by Lemma 2 (Appendix A), with $\theta_2^\star$ as stated right after Equation (6) for both the update rules. Since $\theta_{2,2}^\star = \frac{1}{16} > 0$ under 'training-on-own', resp. $\theta_{2,2}^\star = -\frac{5}{24} < 0$ under 'training-on-all', $\arg\max_{a \in \mathcal{A}_2} \langle \theta_2^\star, x_{2,ta} \rangle$ would fail to select $a_t^\star$ when $X_{(2)}$, resp. $X_{(4)}$, is sampled (see Equation (6)). This would translate into an expected instantaneous regret of at least $\Delta := \min_i \bar{r}_i = \frac{1}{6} > 0$. Hence by Equation (1) and $\mathbb{P}(a_t^\star = a) = |\mathcal{A}|^{-1}$*

$$\mathbb{E}[\mathrm{R}_T^{2s}] \geq \mathbb{E}[\mathrm{R}_T^{\mathbb{N}}] \geq \frac{\Delta}{|\mathcal{A}|} \sum_{a \in \mathcal{A}_2} \mathbb{P}(a_{2,t} \neq a \mid a_{2,t} \in \mathcal{A}_{2,t}^{\mathsf{G}}, a_t^\star = a) \, \mathbb{P}(a_{2,t} \in \mathcal{A}_{2,t}^{\mathsf{G}} \mid a_t^\star = a) \,. \quad (7)$$

*For 'training-on-own', $\mathbb{P}(a_{2,t} \neq a_{(2)} \mid a_{2,t} \in \mathcal{A}_{2,t}^{\mathsf{G}}, a_t^\star = a_{(2)}) = \mathbb{P}(\hat{\theta}_{2,t2} \cdot (-1) < 0) \to 1$ by the above established $\hat{\theta}_{2,t} \to \theta_2^\star$ in probability, and the continuous mapping theorem. Analogously for 'training-on-all'. Hence $\limsup T^{-1} \mathbb{E}[\mathrm{R}_T^{2s}] \geq |\mathcal{A}|^{-1}\Delta\delta > 0$ by Assumption 1, a contradiction, meaning both modes of training fail, but for a different item ($a_{(3)}$ is picked out correctly by both again by the convergence in probability). To make $\limsup T^{-1} \mathbb{E}[\mathrm{R}_T^{\mathbb{N}}] \to 0$ for exactly one of the two setups, add a* third nominator *with $\mathcal{A}_3 = \{a_{(2)}\}$, resp. $\mathcal{A}_3 = \{a_{(4)}\}$, so that $\mathbb{P}(a_t^\star \in \mathcal{C}_t) \to 1$.*

Proposition 2 shows that the nominator training objective can be all the difference between poor and optimal two-stage recommender.[5] Moreover, neither 'training-on-own' nor 'training-on-all' guarantees sublinear regret, and one can fail exactly when the other works. The main culprit is the difference between context distribution in and outside of each pool: combined with the misspecification, either one can result in more favorable optima from the overall two-stage performance perspective. This is the case *both in the supervised and the bandit setting*.

Proposition 2 can be trivially extended to higher number of arms and nominators (add embedding dimensions, let the new arms have non-zero embedding entries only in the new dimensions, and the expected rewards to be lower than the ones we used above). We think that the difference between the in- and out-pool distributions could be exploited to derive analogous results to Proposition 2 for non-linear (e.g., factorization based) models, although the proof complexity may increase.

To summarize, beyond the actual number of nominators identified in the previous section, we have found that the combination of training objective and pool allocation can heavily influence the overall performance. We use these insights to improve two-stage systems in the next section.

## 4 Learning pool allocations with Mixture-of-Experts

Revisiting the proof of Proposition 2, we see the employed pool allocations are essentially adversarial with respect to the context distributions. However, we are typically free to design the pools ourselves, with the only constraints imposed by computational and statistical performance requirements. Proposition 2 thus hints at a positive result: a good pool allocation can help us achieve an (asymptotically) optimal performance *even in cases where this is not possible using any one of the nominators alone.*

Crafting a good pool allocation *manually* may be difficult, and could lead to very bad performance if not done carefully (Proposition 2). We thus propose to *learn* the pool allocation using a **Mixtures-of-Experts** (MoE) [47, 49, 50, 99] based approach instead. A MoE computes predictions by weighting the individual expert (nominator) outputs using a trainable gating mechanism. The weights can be thought of as a *soft pool allocation* which allows each expert to specialize on a different subset of the input space. This makes the MoE more flexible than any one of the experts alone, alleviating the lower modeling flexibility of the nominators due to the latency constraints.

We focus on the Gaussian MoE [47], trained by likelihood maximization (Equation (8)). We employ gradient ascent which—despite its occasional failure to find a good local optimum [68]—is easy to scale to large datasets using a stochastic approximation of gradients

$$\frac{1}{T} \sum_{t=1}^{T} \log \sum_{n=1}^{N} p_{n,t} \exp\left\{ -\frac{(r_t - \hat{r}_{n,t})^2}{2\sigma^2} \right\} \approx \frac{1}{B} \sum_{t=1}^{B} \log \sum_{n=1}^{N} p_{n,t} \exp\left\{ -\frac{(r_t - \hat{r}_{n,t})^2}{2\sigma^2} \right\}, \quad (8)$$

---

[5]Covington et al. [20] reported empirically observing that the the training objective choice has an outsized influence on the performance of two-stage recommender systems. Proposition 2 can be seen as a theoretical complement which shows that the range of important choices goes beyond the selection of the objective.

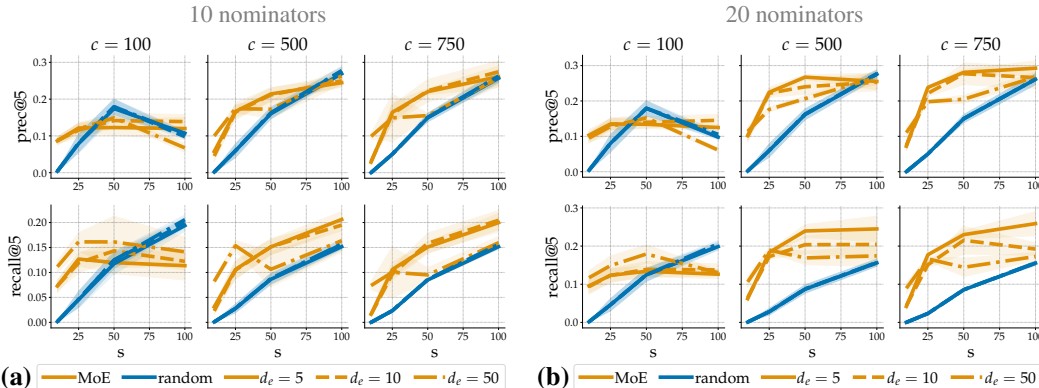

Figure 5: Mixture-of-Experts results on the 100-item Amazon dataset. The x-axis is the size of the BERT embedding dimension subset. The y-axis shows the average `precision@5` (top row) and `recall@5` (bottom row) over 50,000 entries from an independent test set (both set to zero for entries with no positive labels—about 5.5% of the test set). The columns in both plots **(a)** and **(b)** correspond to the number of examples per arm $c$ in the training set. Ten (resp. twenty) nominators were used in **(a)** (resp. **(b)**). The legend shows whether pool allocations were learned (MoE) or randomly assigned (random), and the dimension of item embeddings $d_e$ employed by both model types.

with $p_{n,t} \geq 0$ ($\sum_n p_{n,t} = 1$) the gating weight assigned to expert $n$ on example $t$, $\hat{r}_{n,t}$ the matching expert prediction, $\sigma > 0$ a hyperparameter approximating reward variance, and $B \in \mathbb{N}$ the batch size.

MoE provides a compelling alternative to a policy gradient style approach applied to the joint two-stage policy $\pi^{\texttt{2s}}(a \mid x) = \sum_{a_1,\ldots,a_N} \pi^{\texttt{R}}(a \mid x, \mathcal{C}) \prod_{n=1}^{N} \pi^{\texttt{N}}_n(a_n \mid x)$ as done in [67]. In particular, a significant advantage of the MoE approach is that the sum over the *exponentially many* candidate sets $\mathcal{C} = \{a_1, \ldots, a_N\}$ is replaced by a sum over *only* $N$ experts, which can be either computed exactly or estimated with dramatically smaller variance than in the candidate set case.

There are (at least) two ways of incorporating MoE into existing two-stage recommender deployments:

1. Use a *provisional* gating mechanism, and then distill the pool allocations from the learned weights, e.g., by thresholding the per arm average weight assigned to each nominator, or by restricting the gating network only to the item features. Once pools are divided, nominators and the ranker may be finetuned and deployed *using any existing infrastructure*.

2. Make the gating mechanism *permanent*, either as (i) a replacement for the ranker, or (ii) part of the nominator stage, reweighting the predictions before the candidate pool is generated. This necessitates change of the existing infrastructure but can yield better recommendations.

Unusually for MoEs, we may want to use a different input subset for the gating mechanism and each expert depending on which of the above options is selected. We would like to emphasize that the MoE approach can be used with any *score-based* nominator architecture including but not limited to the linear models of the previous section. If some of the nominators are *not* trainable by gradient descent but *are* score-based, they can be pretrained and then plugged in during the MoE optimization, allowing the other experts to specialize on different items.

We use the 'AmazonCat-13K' dataset [10, 69] to investigate the setup with a logistic gating mechanism as a part of the nominator stage. We employ the same preprocessing as in Section 3.1. Due to the success of greedy methods in Section 3.1, and the existence of black-box reductions from bandit to offline learning [31, 86], we simplify by focusing only on offline evaluation. We compare the MoE against the same model except with the gating replaced by a random pool allocation fixed at the start.

The experts in both models use a simple two-tower architecture, where $d_e$-dimensional dense embeddings are learned for each item, the $s$-dimensional subset of the BERT embeddings is mapped to $\mathbb{R}^{d_e}$ by another trained matrix, and the final prediction is computed as the dot product on $\mathbb{R}^{d_e}$. To enable low latency computation of recommendations, the gating mechanism models the logits $\{\log p_n\}_n$ as a sum of learned user and item embeddings. Further details are described in Appendix B.

Figure 5 shows that MoEs are able to outperform random pool allocation for most combinations of model architecture and training set size. The improved results in recall suggest that the specialization allows nominators to produce a more diverse candidate set. Since *the gating mechanism can learn to exactly recover any fixed pool allocation*, the *MoE can perform worse only when the optimizer fails or the model overfits*. This seems to be happening for the smallest training set size ($c = 100$ samples per arm), and also when the item embedding dimension $d_e$ is high. In practice, these effects can be counteracted by tuning hyperparameters for the specific setting, regularization, or alternative training approaches based on expectation–maximization or tensor decomposition [50, 68, 99].

## 5 Other related work

**Scalable recommender systems.** Interest in scalable recommenders has been driven by the continual growth of available datasets [37, 63, 83, 91]. The two-stage architectures examined in this paper have seen widespread adoption in recommendation [12, 20, 28, 101, 102], and beyond [5, 100]. Our paper is specifically focused on recommender systems which means our insights may not transfer to application areas like information retrieval without adaptation.

**Off-policy learning and evaluation.** Updating the recommendation policy online, without human oversight, runs the risk of compromising the service quality, and introducing unwanted behavior. Offline learning from logged data [27, 73, 89, 92] is an increasingly popular alternative [18, 48, 67, 90]. It has also found applications in search engines, advertising, robotics, and more [4, 48, 62, 88].

**Ensembling and expert advice.** The goal of 'learning with expert advice' [3, 7, 65, 87] is to achieve performance comparable with the best expert if deployed on its own. This is not a good alternative to our MoE approach since two-stage systems typically outperform any one of the nominators alone (Section 3). A better alternative may possibly be found in the literature on 'aggregation of weak learners' [14, 15, 33, 34, 43], or recommender ensembling (see [16] for a recent survey).

## 6 Discussion

We used a combination of empirical and theoretical tools to investigate the differences between single- and two-stage recommenders. Our **first major contribution** is demonstrating that besides common factors like item pool size and model misspecification, the nominator count and training objective can have even larger impact on performance in the two-stage setup. As a consequence, two-stage systems *cannot* be fully understood by studying their components in isolation, and we have shown that the common practice of training each component independently may lead to suboptimal results. The importance of the nominator training inspired our **second major contribution**: identification of a link between two-stage recommenders and Mixture-of-Experts models. Allowing each nominator to specialize on a different subset of the item pool, we were able to significantly improve the two-stage performance. Consequently, splitting items into pools within the nominator stage is not just a way of lowering latency, but can also be used to improve recommendation quality.

Due to the the lack of access, a major limitation of our work is not evaluating on a production system. This may be problematic due to the notorious difficulty of offline evaluation [9, 57, 80]. We further assumed that recommendation performance is captured by a few measurements like regret or precision/recall at $K$, even though design of meaningful evaluation criteria remains a challenge [23, 41, 51, 71]; we caution against deployment without careful analysis of downstream effects and broader impact assessment. Several topics were left to future work: (i) extension of the linear regret proof to non-linear models such as those used in the MoE experiments; (ii) slate (multi-item) recommendation; (iii) theoretical understanding of how much can the ranker reduce the regret compared to the best of the (misspecified) nominators; (iv) alternative ways of integrating MoEs, including explicit distillation of pool allocations from the learned gating weights, learning the optimal number of nominators [77], using categorical likelihood [99], and sparse gating [29, 85].

Overall, we believe better understanding how two-stage recommenders work matters due to the enormous reach of the platforms which employ them. We hope our work inspires further inquiry into two-stage systems in particular, and the increasingly more common 'algorithm-algorithm' interactions between independently trained and deployed learning algorithms more broadly.

## Acknowledgments and Disclosure of Funding

The authors thank Matej Balog, Mateo Rojas-Carulla, and Richard Turner for their useful feedback on early versions of this manuscript. Jiri Hron is supported by an EPSRC and Nokia PhD fellowship.

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
