# A  Auxiliary lemmas

Throughout the paper, we assume the **'stack of rewards model'** from chapter 4.6 of [60].

**Lemma 1** *Let $\hat{\theta}_t = \Sigma_t \sum_{i=1}^{t-1} x_i r_i$ where $\Sigma_t = (\lambda I + \sum_{i=1}^{t-1} x_i x_i^\top)^{-1}$ for some fixed $\lambda \geq 0$. Assume $(x_i, r_i)$ are i.i.d. with $\mathbb{E}[x_1 r_1]$ and $\mathbb{E}[x_1 x_1^\top]$ well-defined, and the latter invertible. Then*

$$\hat{\theta}_t \to (\mathbb{E}[x_1 x_1^\top])^{-1} \mathbb{E}[x_1 r_1] \quad a.s. \tag{9}$$

**Proof 2** *Rewriting $\hat{\theta}_{t+1} = (\frac{\lambda}{t} I + \frac{1}{t} \sum_{i=1}^{t} x_i x_i^\top)^{-1} \frac{1}{t} \sum_{i=1}^{t} x_i r_i$, $\frac{\lambda}{t} I + \frac{1}{t} \sum_{i=1}^{t} x_i x_i^\top \to \mathbb{E}[x_1 x_1^\top]$ and $\frac{1}{t} \sum_{i=1}^{t} x_i r_i \to \mathbb{E}[x_1 r_1]$ a.s. by the strong law of large numbers. Since $A \mapsto A^{-1}$ is continuous on the space of invertible matrices, the result follows by the continuous mapping theorem.*

**Lemma 2** *Consider the setup from Part II of the proof of Proposition 2. Define*

$$\hat{\theta}_{n,t} = \Sigma_{n,t} \sum_{i=1}^{t-1} w_{n,t} x_{n,i} r_i \,, \quad \Sigma_{n,t} = (\lambda I + \sum_{i=1}^{t-1} w_{n,i} x_{n,i} x_{n,i}^\top)^{-1} \,,$$

*with $\lambda \geq 0$ fixed, and $\theta_n^\star = (\mathbb{E}[w_{n,a} x_{n,a} x_{n,a}^\top])^{-1} \mathbb{E}[w_{n,a} x_{n,a} r_a]$, $a \sim Unif(\mathcal{A})$. Then $\hat{\theta}_{n,t} \to \theta_n^\star$ in probability, if $\limsup T^{-1} \mathbb{E}[R_T^{2s}] \to 0$.*

**Proof 3** *Since $x_{n,t} = 0$ unless $a_t = a_t^\star$ by construction, $\hat{\theta}_{n,t+1}$ is equal to*

$$\left( \frac{\lambda}{t} I + \frac{1}{t} \sum_{i=1}^{t} \sum_{j=1}^{|\mathcal{A}|} \mathbb{1}\{a_i^\star = a_i = a_{(j)}\} w_{n,i} x_{n,i} x_{n,i}^\top \right)^{-1} \frac{1}{t} \sum_{i=1}^{t} \sum_{j=1}^{|\mathcal{A}|} \mathbb{1}\{a_i^\star = a_i = a_{(j)}\} w_{n,i} x_{n,i} r_i \,.$$

*Define $S_{t(j)}^\star := \sum_{i=1}^{t-1} \mathbb{1}\{a_i^\star = a_{(j)}\} w_{n,i}$ for each $a_{(j)} \in \mathcal{A}$, and take, for example, the term*

$$\sum_{j=1}^{|\mathcal{A}|} \frac{S_{t(j)}^\star}{t-1} \frac{1}{S_{t(j)}^\star} \sum_{i=1}^{t-1} \mathbb{1}\{a_i^\star = a_i = a_{(j)}\} w_{n,i} x_{n,i} r_i \,.$$

*Since $a_t^\star \overset{i.i.d.}{\sim} Unif(\mathcal{A})$ by construction, $\frac{S_{t(j)}^\star}{t-1} \to |\mathcal{A}|^{-1}$ a.s. by the strong law of large numbers, and $S_{t(j)} \to \infty$ a.s. by the second Borel-Cantelli lemma. Furthermore, defining $S_{t(j)} := \sum_{i=1}^{t-1} \mathbb{1}\{a_i^\star = a_i = a_{(j)}\} w_{n,i}$, $S_{t(j)}^\star - S_{t(j)} \geq 0$ is the number of '$a_{(j)}$ mistakes', and is associated with positive regret when the inequality is strict. Observe that we must have $t^{-1}(S_{t(j)}^\star - S_{t(j)}) \to 0$ in probability, as otherwise there would be $c, \epsilon > 0$ such that $\limsup \mathbb{P}(t^{-1}(S_{t(j)}^\star - S_{t(j)}) > \epsilon) > c$, implying*

$$\limsup T^{-1} \mathbb{E}[R_T^{2s}] \geq \limsup T^{-1} \mathbb{E}[R_T^N] \geq \Delta \limsup \mathbb{E}\left[ \frac{S_{T(j)}^\star - S_{T(j)}}{T} \right] > \Delta c \epsilon > 0 \,,$$

*which contradicts the assumption $\limsup T^{-1} \mathbb{E}[R_T^{2s}] \to 0$ (recall $\Delta = \min_i \bar{r}_i > 0$).*

*Finally, $t^{-1}(S_{t(j)}^\star - S_{t(j)}) \to 0$ implies $\frac{S_{t(j)}}{S_{t(j)}^\star} \to 1$ and $S_{t(j)} \to \infty$ in probability, and therefore*

$$\frac{S_{t(j)}^\star}{t-1} \sum_{j=1}^{|\mathcal{A}|} \frac{S_{t(j)}}{S_{t(j)}^\star} \frac{1}{S_{t(j)}} \sum_{i=1}^{t-1} \mathbb{1}\{a_i^\star = a_i = a_{(j)}\} w_{n,i} x_{n,i} r_i \to \mathbb{E}_{a \sim Unif(\mathcal{A})}[w_{n,a} x_{n,a} r_a] \,,$$

*in probability by the law of large numbers, the continuous mapping theorem, and $|\mathcal{A}| < \infty$. Since an analogous argument can be made for the covariance term, and $A \mapsto A^{-1}$ is continuous on the space of invertible matrices, $\hat{\theta}_{n,t} \to \theta_n^\star$ in probability by the continuous mapping theorem, as desired.*

# B  Experimental details

The experiments were implemented in Python [93], using the following packages: abseil-py, h5py, HuggingFace Transformers [97], JAX [13], Jupyter [55], matplotlib [45], numpy [39], Pandas [74, 96],

PyTorch [75], scikit-learn [76], scipy [94], seaborn [95], tqdm. The bandit experiments in Section 3 were run in an embarrassingly parallel fashion on an internal academic CPU cluster running CentOS and Python 3.8.3. The MoE experiments in Section 4 were run on a single desktop GPU (Nvidia GeForce GTX 1080). While each experiment took under five minutes (most under two), we evaluated hundreds of thousands of different parameter configurations (including random seeds in the count) over the course of this work. Due to internal scheduling via slurm and the parallel execution, we cannot determine the overall total CPU hours consumed for the experiments in this work.

Besides the UCB and Greedy results reported in the main text, some of the experiments we ran also included *policy gradient* (PG) where at each step $t$, the agent takes a single gradient step along $\nabla \mathbb{E}_x \mathbb{E}_{a \sim \pi(x)} r_a = \mathbb{E}_x \mathbb{E}_{a \sim \pi(x)} r_a \nabla \log \pi_a(x)$ where the policy is parametrised by logistic regression, i.e., $\log \pi_a(x) = \langle \theta, x_a \rangle - \log \sum_{a'} \exp\{\langle \theta, x_{a'} \rangle\}$, and the expectations are approximated with the *last* observed tuple $(x_t, a_t, r_t)$. PG typically performs much worse than UCB and Greedy in our experiments which is most likely the result of not using a replay buffer, or any of the other standard ways of improving PG performance. We eventually decided not include the PG results in the main paper as they are not covered by the theoretical investigation in Section 3.1.

For the bandit experiments, the arm pools $\{\mathcal{A}_n\}_n$, and feature subsets $s < d$, were divided to minimize overlaps between the individual nominators. The corresponding code can be found in the methods `get_random_pools` and `get_random_features` within `run.py` of the supplied code:

- **Pool allocation:** Arms are randomly permuted and divided into $N$ pools of size $\lfloor |\mathcal{A}|/N \rfloor$ (floor). Any remaining arms are divided one by one to the first $|\mathcal{A}| - N \lfloor |\mathcal{A}|/N \rfloor$ nominators.
- **Feature allocation:** Features are randomly permuted and divided into $N$ sets of size $s' = \min\{s, \lfloor d/N \rfloor\}$. If $s' < s$, the $s - s'$ remaining features are chosen uniformly at random without replacement from the $d - s'$ features not already selected.

To adjust for the varying dimensionality, the regularizer $\lambda$ was multiplied by the input dimension for UCB and Greedy algorithms, throughout. The $\lambda$ values reported below are prior to this scaling.

## B.1 Synthetic bandit experiments (Figure 3)

**Hyperparameter sweep:** We used the single-stage setup, no misspecification ($d = s$), 100 arms, $d = 20$ features, and 0.1 reward standard deviation, to select hyperparameters from the grid in Table 1, based on the average regret at $T = 1000$ rounds estimated using 30 different random seeds.

Table 1: Hyperparameter grid for the synthetic dataset. Bold font shows the selected hyperparameters.

| algorithm | parameter | values |
|---|---|---|
| UCB | regularizer $\lambda$ | $[10^{-4}, 10^{-3}, \mathbf{10^{-2}}, 10^{-1}, 10^{0}, 10^{1}, 10^{2}]$ |
|  | exploration bonus $\alpha$ | $[10^{-4}, 10^{-3}, \mathbf{10^{-2}}, 10^{-1}, 10^{0}, 10^{1}, 10^{2}]$ |
| Greedy | regularizer $\lambda$ | $[10^{-4}, 10^{-3}, \mathbf{10^{-2}}, 10^{-1}, 10^{0}, 10^{1}, 10^{2}]$ |
| PG | learning rate | $[10^{-4}, 10^{-3}, 10^{-2}, 10^{-1}, \mathbf{10^{0}}, 10^{1}, 10^{2}]$ |

With the hyperparameters fixed, we ran 30 independent experiments for each configuration of the UCB, Greedy, and PG algorithms in the single-stage case, and 'UCB+UCB', 'UCB+PG', 'UCB+Greedy', 'PG+PG', and 'Greedy+Greedy' in the two-stage one. Other settings we varied are in Table 2. The 'misspecification' $\rho$ was translated into the nominator feature dimension via $s = \lfloor d/\rho \rfloor$. For the nominator count $N$, the configurations with $N > |\mathcal{A}|$ were not evaluated.

Table 2: Evaluated configurations for the synthetic dataset.

| parameter | values |
|---|---|
| arm count $|\mathcal{A}|$ | $[10, 100, 10000]$ |
| feature count $d$ | $[5, 10, 20, 40, 80]$ |
| nominator count $N$ | $[2, 5, 10, 20]$ |
| reward std. deviation | $[0.01, 0.1, 1.0]$ |
| misspecification $\rho$ | $[0.2, 0.4, 0.6, 0.8, 1.0]$ |

## B.2 Amazon bandit experiments (Figure 4)

The features were standardized by computing the mean and standard deviation over *all* dimensions.

**Hyperparameter sweep:** We used the single-stage setup, $s = 50$ features, 100 arms, to select hyperparameters from the grid in Table 3, based on the average regret at $T = 5000$ rounds estimates using 30 different random seeds.

Table 3: Hyperparameter grid for the Amazon dataset. Bold font shows the selected hyperparameters.

| algorithm | parameter | values |
|-----------|-----------|--------|
| UCB | regularizer $\lambda$ | $[10^{-4}, 10^{-3}, 10^{-2}, 10^{-1}, \mathbf{10^0}, 10^1, 10^2]$ |
| | exploration bonus $\alpha$ | $[10^{-4}, \mathbf{10^{-3}}, 10^{-2}, 10^{-1}, 10^0, 10^1, 10^2]$ |
| Greedy | regularizer $\lambda$ | $[10^{-4}, 10^{-3}, 10^{-2}, 10^{-1}, \mathbf{10^0}, 10^1, 10^2]$ |
| PG | learning rate | $[10^{-4}, 10^{-3}, 10^{-2}, 10^{-1}, 10^0, \mathbf{10^1}, 10^2]$ |

With the hyperparameters fixed, we again ran 30 independent experiments for the same set of algorithms as in Appendix B.1, but now with fixed $d = 400$ as described in Section 3.1. Since $d$ is fixed, we vary the nominator feature dimension $s$ directly. Other variables are described in Table 4.

Table 4: Evaluated configurations for the Amazon dataset.

| parameter | values |
|-----------|--------|
| arm count $|\mathcal{A}|$ | $[10, 100, 1000]$ |
| nominator count $N$ | $[2, 5, 10, 20]$ |
| nominator feature dim $s$ | $[5, 10, 20, 40, 80, 150, 250, 400]$ |

## B.3 Mixture-of-Experts offline experiments (Section 4)

**Hyperparameter sweep:** We ran a separate sweep for the MoE and the random pool models using 100 arms, $N = 10$ experts, and $c_k = 500$ training examples per arm. We swept over optimizer type ('RMSProp' [42], 'Adam' [54]), learning rate ($[0.001, 0.01]$), and likelihood variance $\sigma^2$ ($[0.01, 1.0]$). The selection was made based on average performance over three distinct random seeds. The 0.01 learning rate was best for both models. 'RMSProp' and $\sigma^2 = 1$ were the best for the random pool model, whereas 'Adam' and $\sigma^2 = 0.01$ worked better for the MoE, except for the embedding dimension $d_e = 50$ where $\sigma^2 = 1$ had to be used to prevent massive overfitting.

**Evaluation:** We varied the number of training examples per arm $c_k \in \{100, 500, 750\}$, number of dimensions of the BERT embedding revealed to the nominators $s \in \{10, 25, 50, 100\}$, the dimension of the learned item embeddings $d_e \in \{5, 10, 50\}$, and the number of experts $N \in \{5, 10, 20\}$. We used 50000 optimization steps, batch size of 4096 to adjust for the scarcity of positive labels, and no early stopping. Three random seeds were used to estimate the reported mean and standard errors.

# C   Additional results

## C.1   Synthetic bandit experiments (Figure 3)

The interpretation of all axes and the legend is analogous to that in Figure 3, except the relative regret (divided by that of the uniformly guessing agent) is reported as in Figure 4.

## C.2   Amazon bandit experiments (Figure 4)

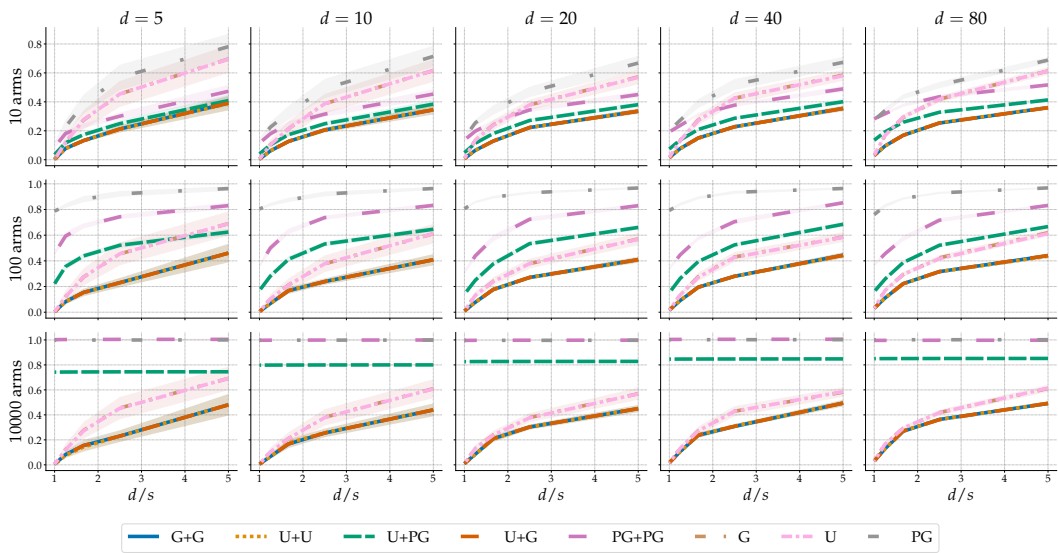

Figure 6: Relative regret for *two* nominators on the synthetic dataset.

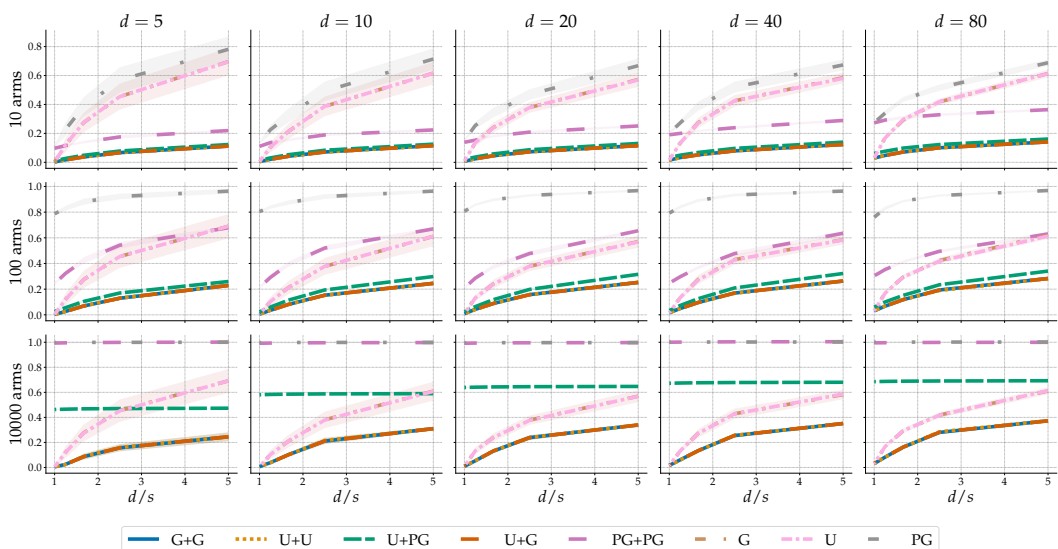

Figure 7: Relative regret for *five* nominators on the synthetic dataset.

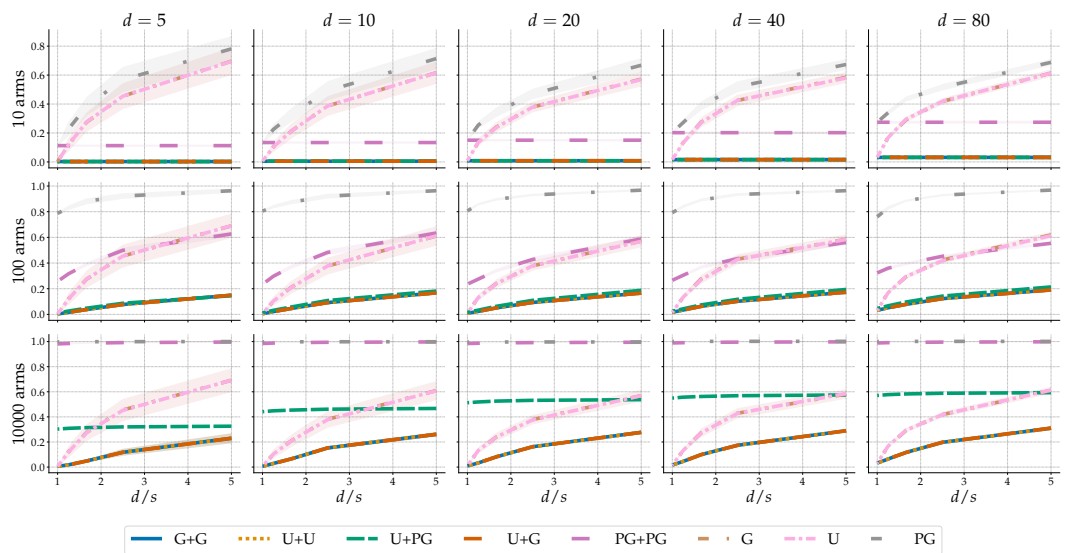

Figure 8: Relative regret for *ten* nominators on the synthetic dataset.

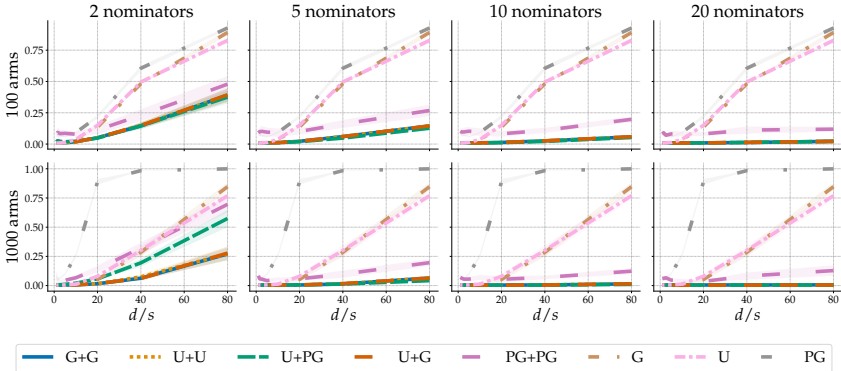

Figure 9: Relative regret on the Amazon dataset.