# OpenReview forum: "On Component Interactions in Two-Stage Recommender Systems"
_NeurIPS.cc/2021/Conference — NeurIPS 2021 Poster_

### Official Review · Reviewer_jFPh · 2021-07-15

**Rating:** 6
**Confidence:** 4

**Summary:**

This work compares one-stage and two-stage systems from both of empirical and theoretical observations, which finds that interactions between the ranker and the nominators substantially affect the overall system performance. A Mixture-of-Experts approach is adopted to train the nominators to specialize on different subsets of the item pool. This improves the performance conducted on a Amazon review dataset. Overall, this submission is well presented and easy to follow.

**Limitations And Societal Impact:**

Yes.

**Main Review:**

Strength:

1.This work targets at looking for two-stage specific properties, an interesting topic for the audience in the area of online large scale information system.

2.Empirical and theoretical observations well motivate its research problem.

3.Literature review is well embedded in the body of this submission.

Weak:

1.Since this work focus on recommender system. My main concern is what are the differences between RS and others. The mentioned ranker and nominator are also existed in some information systems like Q&A, Web search, sentence matching, and so on.

2.Without clearly addressing those differences, readers may feel that the current version can be applied mechanically to other IS.

3.Back to RS, there are various benchmark datasets. The reason of selecting “AmazonCat-13K” is not known. Experiments are conducted on review texts, so if only using rating scores, does it have different effects on the current findings?

4.The way to define two-stage RS is based on the bandit based approach. The reason and its advantages should be further addressed.

**Time Spent Reviewing:**

6

---

> ### Author Response · Authors · 2021-08-10
> **Response to jFPh**
>
> > “My main concern is what are the differences between RS and others. The mentioned ranker and nominator are also existed in some information systems like Q&A, Web search, sentence matching, … Without clearly addressing those differences, readers may feel that the current version can be applied mechanically to other IS.”
>
> This is a very good point as the fact that two-stage systems are being used beyond recommender systems is only very briefly mentioned in Section 5, without any discussion of the differences. We will extend our discussion in the next revision of the paper, and include a clear acknowledgement that our proposals need adaptation—or may not be applicable (e.g., if there is no nominator trainable by minimisation of a weighted objective like MSE)—if they are to be used in the other application areas. If there are any particular papers or topics you would recommend, we would be more than happy to incorporate them!
>
> > “ The reason of selecting “AmazonCat-13K” is not known. Experiments are conducted on review texts, so if only using rating scores, does it have different effects on the current findings?”
>
> The deployments of large scale two-stage recommender systems we are familiar with strongly (or even exclusively) depend on contextual information about the user/query (e.g., [1] and the follow-up work). If the model depended only on user and item IDs which do not change when deployed, recommendations for each user could be pre-computed and retrieved in constant time at deployment time, obviating the need for using a computationally efficient architecture like the two-stage one we study (with approximate nearest neighbour search at deployment). Please let us know if your experience/understanding is different, we would be keen to discuss!
>
> In light of the above, we ruled out the MovieLens dataset where the contextual information (occupation, sex, age, etc.) is too general to carry much useful information. This is evidenced by the fact that even state-of-the-art MovieLens models ignore the contextual information [2]. Beyond MovieLens, there are several options, and we picked AmazonCat because it is both reasonably popular and comes with contextual information that is useful for the prediction task. Of course, the best option would have been to evaluate in deployment but we unfortunately do not have access to any industrially deployed system (lines 314-315). We will emphasise this limitation more in the next revision.
>
> > “The way to define two-stage RS is based on the bandit based approach. The reason and its advantages should be further addressed.”
>
> Since we are not 100% sure, could you please clarify what is meant by the “bandit based approach to defining two-stage RS” here? If it is that we use score-based ranking, this is an approach that is quite popular in two-stage recommenders (e.g., [1] and the follow-up work), but definitely not the only one—more on this in the response to Q3hm’s question about the alternative Learning to Rank approaches. If what is meant is that our theoretical analysis is focused on regret based loss and simple linear models, perhaps our general answer to all reviewers at the top is adequate? If something else was meant, please let us know, we would be more than happy to discuss!
>
> REFERENCES:
>
> [1] Covington, Adams, Sargin. Deep Neural Networks for YouTube Recommendations. 2016.
>
> [2] Rendle, Steffen, Li Zhang, and Yehuda Koren. "On the difficulty of evaluating baselines: A study on recommender systems." arXiv preprint arXiv:1905.01395 (2019).

---

### Official Review · Reviewer_Q3hM · 2021-07-15

**Rating:** 5
**Confidence:** 3

**Summary:**

In this paper, the authors are trying to optimize two stage recommendation systems (first stage to generate candidates and second stage to rank the candidates for a given context).

**Limitations And Societal Impact:**

The authors have identified no limitations or  potential negative impact.

Potential Limitation :
One of the main reasons for popularity of 2 stage recommendations is the scalability & flexibility they provide for large scale recommender systems (independence of creating a candidate set and a separate ranker). Increasing the number of nominators & MOE model adds more complexity and increases computational cost for the proposed 2-stage recommender system by the authors.


**Main Review:**

The idea of being able to do combined optimization on the rankers and candidate generators together is extremely useful. The authors proposed solution to use MOE model to create multiple candidate/item pool (nominators) specialized on different sub-tasks and final ranker to select the best candidate can find a lot of applications.

The authors have given extensive explanation with solid theoretical proof for their proposed solution. They have presented a thorough study of different components in their system and how it effects the final stage performance of the ranker.

The evaluations presented in the paper support the author’s claims and observations. However, the paper is not extremely easy to read especially the experimental setup could have been better explained, and I have listed down some opportunities the authors could have presented better.

Opportunities:
1.	There is no evaluation against any Learning to rank models, extremely popular with the two stage recommendation systems. Their main performance comparison is against single stage performance, but the setup for that evaluation/theoretical justification remains the same (EE based system).
2.	The related work section can be better explored.

**Time Spent Reviewing:**

5

---

> ### Author Response · Authors · 2021-08-10
> **Response to Q3hM**
>
> > “​​the paper is not extremely easy to read especially the experimental setup could have been better explained”
>
> We will be incorporating feedback from the other reviewers into our next revision. However, if you have any suggestion regarding clarity not mentioned by others, we would be happy to incorporate them.
>
> > “There is no evaluation against any Learning to rank models”
>
> The focus of this paper is on evaluating the effects of different two-stage setups, not the quality of different predictive models. Since we are aware of only one two-stage learning-to-rank paper [1] (which is focused on document retrieval, not recommendation), we purposefully restricted our evaluation to score-based models for the nominators/rankers (lines 58-59) as is common in prior work (e.g., [2, 3, 4]). We did this for three reasons:
> 1. There are an intractable number of models for us to compare. By restricting the set of nominator/ranker models under consideration, we are able to perform a more in-depth comparison across different two-stage setups. Furthermore, by using only simpler score-based models, we are able to strip away performance disparities that could originate from differences in each model’s predictive power, rather than the two-stage setup. (More on this in our general response to all reviewers at the top.)
> 2. The canonical L2R models are not comparable on the tasks we consider. While there are some exceptions, most of the popular L2R models are developed and evaluated for settings where the model observes an explicit score for each item in a query [5, 6, 7, 8], while in the tasks we consider, the model only observes implicit feedback for the item they recommend. Adding tasks that would allow us to directly compare against L2R would have added complexity, while not addressing our core research question.
> 3. Score-based ranking approaches are at least as popular as L2R models (e.g., [2, 3, 4]), especially in the nominator stage of large-scale systems as they enable computationally efficient generation of candidates by approximate nearest neighbour search (e.g., [2]).
>
> We do agree that an extension to L2R would be very interesting, but this is out of scope of this work (lines 319-320). We will advertise the restriction to score-based ranking models more clearly in the introduction/abstract of the next revision.
>
> > “The related work section can be better explored.”
>
> We do agree that we could have included Learning to rank as related work, which will be remedied in the next revision (if there are any key references to L2R based two-stage systems we missed, please let us know!).
>
> > “The authors have identified no limitations or potential negative impact.”
>
> We respectfully disagree. In Section 6 (lines 314-323) we identify several key limitations of our work and opportunities for future work (including slate based recommendations). We also included a discussion of potential impacts both in our introduction (lines 32-40), and our discussion (lines 324-327), which the other reviewers found sufficient. We are of course open to any specific suggestions you may have.
>
> > “Increasing the number of nominators & MOE model adds more complexity and increases computational cost for the proposed 2-stage recommender system by the authors.”
>
> We do not require increasing the number of nominators anywhere in the paper. Instead, our proposals apply and can be used to improve any system with two or more nominators. For the MoE, we explicitly comment on how to use the MoE algorithm only for pre-training, followed by reverting to standard architecture on top of the learned pools (1st item in the numbered list within Section 4, lines 259-263). Discussion of other options of employing the MoE and how to design the gating network in a two-tower like way are also described in Section 4.
>
> That being said, adding more nominators can sometimes improve latency rather than worsen it, as separate nominators can be easily parallelized across multiple machines, each operating only on its own subset of the item pool. The gained latency can then of course be traded-off for greater modelling capacity of the nominators.
>
> Please let us know if you have any remaining concerns, we would be happy to discuss! We would particularly appreciate if you could please highlight steps we could take to increase our score, if any.
>
> REFERENCES:
>
> [1] Dang, Bendersky, Croft. Two-stage learning to rank for information retrieval. 2013.
>
> [2] Covington, Adams, Sargin. Deep Neural Networks for YouTube Recommendations. 2016.
>
> [3] Chen, Beutel, Covington, Jain, Belletti, Chi. Top-K off-policy correction for a REINFORCE recommender system. 2019.
>
> [4] Ma, Zhao, Yi, Yang, Chen, Tang, Hong, Chi. Off-policy learning in two-stage recommender systems. 2020.
>
> [5] Chapelle, Olivier, and Yi Chang. Yahoo! learning to rank challenge overview. 2011.
>
> [6] Qin, Tao, and Tie-Yan Liu. Introducing LETOR 4.0 datasets. 2013.
>
> [7] Burges, Christopher, Robert Ragno, and Quoc Le. Learning to rank with nonsmooth cost functions. 2006.
>
> [8] Burges, Chris, Tal Shaked, Erin Renshaw, Ari Lazier, Matt Deeds, Nicole Hamilton, and Greg Hullender. Learning to rank using gradient descent. 2005.

---

### Official Review · Reviewer_GChu · 2021-07-17

**Rating:** 5
**Confidence:** 4

**Summary:**

This paper studies on the properties of two-stage recommendation architecture. Both empirical and theoretical analyses have been conducted to reveal the impact of pool allocation on recommendation performance. The authors further propose a MOE-based learning strategy for pool allocation, which yields empirical improvement over the random pool allocation.

**Limitations And Societal Impact:**

Yes

**Main Review:**

Strengths:
1. This paper studies on the properties of the two-stage recommendation architecture. It is an interesting and under-explored topic.
2. Both empirical and theoretical analyses are conducted to validate the importance of pool allocation on recommendation performance.
3. The authors further propose to learn pool allocation with MOE strategy, which yields empirical improvement over the random pool allocation.

Weaknesses:

1. The analyses are based on some strong assumptions, which may not be consistent with the setting of real-world two-stage recommendation system.

For example, in real-world RS, we usually use complex model (e.g. MF, FM, DeepFM) instead of simple linear model; Also, each nominator may return a list of items rather than single item; What’s more serious, the assigned pool of nominators in real-world RS usually covers all items (ie. A_n=A). I do not understand why we need pool allocation in this scenario.

2. The proof of Proposition 2 is not complete.
The proof is on the specific case with two nominators and three items, which lacks generalization.

3. Some important concepts lack clear interpretation, making the paper hard to follow.
For example, what does “context distribution” mean? Does it mean \pi(x), or mean the distribution of x over the items in the pool? Also, in figure 2, what does “ G” or “U” mean? Does it mean one-stage recommender?

4. Insufficient motivation of using MOE.

We do agree that learning pool allocation is promising. But whether MOE can learn a proper allocation is questionable. More concretely, we do not understand why the weights returned by the MOE is a useful signal for pool allocation. The authors should give more deep analyses on MOE and explain why MOE is so special for pool allocation.


In summary, although this paper studies an interesting and important problem, it has some serious limitations in terms of presentation and theoretical soundness. As such, I think this paper is potential but this version is not touch the bar of NIPS.


**Time Spent Reviewing:**

4hours

---

> ### Author Response · Authors · 2021-08-10
> **Response to GChu**
>
> > “The analyses are based on some strong assumptions, which may not be consistent with the setting of real-world …”
>
> >> “In real-world RS, we usually use complex model (e.g. MF, FM, DeepFM) instead of simple linear model … each nominator may return a list of items rather than single item”
>
> We agree this is a limitation. Please see our general answer to all reviewers at the top. The reasoning behind single-item vs. slate recommendation is analogous, and we mention it as a limitation and subject of future work (lines 58-59 and 319-320). That being said, the two-tower architecture used within the experimental part of Section 4 is part of some of the largest known deployments of two-stage recommendation systems [1]. We will clarify this in the next revision.
>
> >> “... the assigned pool of nominators in real-world RS usually covers all items (ie. A_n=A). I do not understand why we need pool allocation in this scenario.”
>
> We respectfully disagree. Firstly, at least one of the currently most prominent two-stage recommender deployments—the YouTube homepage algorithm—is using distinct item pools to achieve improved scalability & item diversity (this is alluded to in section 2 of [1], and the related talk [2]; an item pool can be constituted from, e.g., only recently uploaded videos). Secondly and more importantly, one of the main insights of our paper is that splitting items into distinct pools can be effectively exploited to improve the average quality of the candidate set by taking the novel view of nominators as a “collection of weak learners” whose predictions can be aggregated in a statistically efficient way (Section 4). This can (partially) compensate for the lower modelling capacity of the nominators by requiring each to only work well for a subset of the whole item pool. We will emphasise this more in the next revision.
>
> > “The proof of Proposition 2 is not complete. The proof is on the specific case with two nominators and three items, which lacks generalization.”
>
> This proof of Proposition 2 *is* complete. We state "There exist two distinct context distributions with associated pool allocations $\lbrace\mathcal{A}_n\rbrace_n$, and reward distributions where $r_a \in [0, 1]$ …” , i.e., all we claim is that there are two problem instances for which the claims hold, not that there exists two instances for any possible combination of the item and ranker counts. This constructive way of proving generalisation lower bounds is standard (e.g. [3]), and has the additional benefit of allowing clearer exposition. That being said, extension of Proposition 2 to more than two nominators and three items is straightforward; on a high-level: (i) extend the ranker dimension by one or more, let the feature vectors for the new items (actions) have non-zero entries only in these new dimensions, and expected rewards lower than any of the arms we used in the original examples; and (ii) let the new nominators have access only to these new actions. We will add a fully rigorous description of this extension into the appendix.
>
> > “Some important concepts lack clear interpretation …”
>
> We will add the following clarifications:
>
> >> “What does “context distribution” mean?”
>
> It is the distribution over the contexts $x$. $\pi(x)$ is the distribution over recommendation (actions) given a context $x$.
>
> >> “In figure 2, what does “ G” or “U” mean? Does it mean one-stage recommender?”
>
> Yes.
>
> > “Insufficient motivation of using MOE.”
>
> >> “... whether MOE can learn a proper allocation is questionable.”
>
> MoE can in theory fit any pool allocation one could design by hand, including the one where all pools are equal if this is indeed optimal from the performance perspective. While it is true that provably finding global optima in MoEs is an area of active research as mentioned in Section 4 (to be fair, this is also the case for most NN-based recommender systems), MoEs have been successfully used in a variety of contexts (e.g., [4]), and the results in Section 4 suggest they also have potential in our application.
>
> >> “... why the weights returned by the MOE is a useful signal for pool allocation. The authors should give more deep analyses on MOE and explain why MOE is so special for pool allocation.”
>
> We cite several alternatives to the MoE approach in the “Ensembling and expert advice” paragraph of Section 5, and thus do not claim MoEs are “special”. Rather, we view our observation that candidate generation can be significantly improved by better pool allocation—which can be achieved by recognising the link to the literature on aggregation of “weak learners” (MoEs, boosting, bagging, etc.)—as the core contribution here.
>
> That being said, MoEs were specifically designed as a way to efficiently aggregate multiple models (experts) by allowing each to specialise on a different subset of the input space, which is exactly what we are trying to achieve. As for “why the weights returned by the MOE is a useful signal for pool allocation”, one can gain intuition by differentiating the objective in Eq. 8 w.r.t. the parameters of a particular nominator, and observing that the gradient is equivalent to that of a weighted MSE objective. The weights are based on the gating network and the relative quality of each prediction compared to the other nominators (experts), which means that the nominator can focus only on the items where it is competitive. This facilitates specialisation. Inspecting the gating network gradients, one sees that assigning probability one to the best performing nominator (expert) is optimal. This encourages sparse pool allocations. As commented on in the numbered list within Section 4, the gating network can be further constrained to satisfy application specific requirements.
>
> > “​​In summary, although this paper studies an interesting and important problem, it has some serious limitations in terms of presentation and theoretical soundness.”
>
> Since the only comment concerning theory was 2., which we addressed above, we do not feel there are “serious limitations” in terms of theoretical soundness. As for the presentation, while this is a matter of taste to a degree (jFPh wrote “Overall, this submission is well presented and easy to follow.”), we will incorporate your feedback and are open to any further suggestions.
>
> Please let us know if you have any remaining concerns, we would be happy to discuss! We would particularly appreciate if you could please highlight steps we could take to increase our score, if any.
>
> REFERENCES:
>
> [1] Covington, Adams, Sargin. Deep Neural Networks for YouTube Recommendations. 2016.
>
> [2] https://youtu.be/WK_Nr4tUtl8?t=166
>
> [3] Lattimore, Szepesvári. Bandit algorithms. 2020.
>
> [4] Yuksel, Wilson, Gader, Twenty Years of Mixture of Experts. 2012.

---

### Official Review · Reviewer_hgnQ · 2021-07-20

**Rating:** 7
**Confidence:** 3

**Summary:**

The work focuses on a very practical problem of designing a good a two-stage recommender systems. The authors shed light on interaction between the two stages (called nominator and ranker) and its effects on an overall performance of the solution. They theoretically prove that the way nominators are trained may significantly affect the end quality and provide an empirical evidence for their findings as well. The authors also provide a recipe for designing a better first stage based on a mixture-of-experts approach that turns out to advantageous over naïve approach in certain cases.


**Limitations And Societal Impact:**

The authors briefly mention connections of recommender systems to some of the issues like biases, fairness, etc. and do not explore it further. However, even for single-stage recommenders this is a very challenging topic. Given that the problem of two-stage recommender systems is underexplored, I believe that putting too much focus on societal and ethical concerns in the presented work wouldn't be helpful. The demonstrated awareness of the problem is enough in this case.

**Main Review:**

I totally agree with the authors that there's a lack of research devoted to two-stage recommenders. Indeed, there are many reports from different companies on the use-cases where such systems provide more advantages over standalone monolithic solutions. This observation is also supported by various recsys challenges where two-stage solutions are also demonstrated to provide highly competitive results. Hence, attempts to better understand their internals and working mechanics are more than welcomed. The work provides good theoretical grounds for the considered cases within the specified constraints and nicely connects it to experimental part.

It should be noted though, the focus is made specifically on bandits-based solutions, which is not the only possible setup. It would be great to see (at least at high level) some connections to other approaches, e.g. when ranker is based on Decision Trees and nominator is based on matrix factorization, which is a very popular setup. Surely, this may lead to many difficulties with analysis and may not even give a way for proper theoretical investigation, but it would be helpful to at least specify the limitations. Nevertheless, the lack of such analysis doesn't decrease the value of the provided results.

Related to the previous comment, an abstract creates an impression of a general solution, while only the case of bandits is considered. I'd suggest to modify the abstract and the introductory part of the text to more clearly position the work and define its boundaries.

I also have a few concerns regarding the experimental part and the assumptions made there. In Section 3.1, the synthetic dataset is generated under the assumption that context follows normal distribution. However, in almost all practical cases, items in recommender systems follow power law or zipf-like distributions, which, I believe, would contradict to the described assumption. It's not clear how this would affect further derivations and provided intuitions. Describing how different assumptions on the nature of contexts may affect the theoretical and/or experimental results of this work would help to better understand its practicality.

The data preprocessing part also seems to deviate form real scenarios. E.g., in Section 4 (lines 273-275): why sampling based on label frequency is performed? How does it affect distribution of data? How it relates to the assumptions on data used in theoretical part?  What is the connection to real world datasets? Please, add more details on the reasons for the described preprocessing.

**Time Spent Reviewing:**

6

---

> ### Author Response · Authors · 2021-08-10
> **Response to hgnQ**
>
> > “the focus is made specifically on bandits-based solutions, which is not the only possible setup. It would be great to see (at least at high level) some connections to other approaches …”
>
> We agree this is not the only possible setup and address this concern in our general reply to all reviewers above. We will acknowledge this focus and its advantages & disadvantages directly in the abstract/introduction as suggested, and emphasise them more throughout the paper.
>
> As for the connection to other approaches, we would intuitively expect that the disparity between the “pool conditional” and “joint across pools'' context distributions—which is at the heart of Proposition 2—could be exploited to construct counterexamples for other algorithms as well, although the complexity of the construction can definitely increase. As mentioned in the general response, Section 4 provides early evidence that our observations do generalise beyond measurements of regret and linear models. We will emphasise this more clearly in the next revision together with a clear acknowledgement that while the results on importance of pool allocations from Section 3 directly motivate the MoE approach in Section 4, further work is necessary to understand the observed behaviour theoretically.
>
> > “In Section 3.1, the synthetic dataset is generated under the assumption that context follows normal distribution. However, in almost all practical cases, items in recommender systems follow power law or zipf-like distributions …”
>
> We agree the setting of the exploratory synthetic experiments may fail to capture properties that may have a significant effect in particular scenarios such as the power law behaviour you mention, temporal and network effects, feedback loops, etc. We believe that gaining understanding of these more complex scenarios is hard without having a grasp on the behaviour in simpler cases such as the one we study. Beyond analytical tractability, an additional benefit of our setup is that it conforms to common assumptions in the bandit literature which allows us to invoke existing results in order to investigate one of the main questions of Section 3, i.e., to what extent do factors known to explain “single-stage behaviour” also affect two-stage systems (within the bandit style setup).
>
> The theoretical results do not require the context distribution to be Gaussian, i.e., are not affected: Proposition 1 holds for any (even adversarial) contexts, and Proposition 2 constructs specific context distributions which serve as counterexamples to superiority of either of the considered training objectives. The counterexamples are simple, but could be potentially embedded in more complicated setups as we describe in response to GChu’s second question. We will add a clarification with clearly delineated limits of the synthetic experiments as you suggest, and hope to extend to some of the more complex scenarios in the future.
>
> > “Section 4 (lines 273-275): why sampling based on label frequency is performed? How does it affect distribution of data? How it relates to the assumptions on data used in theoretical part? What is the connection to real world datasets? Please, add more details on the reasons for the described preprocessing.”
>
> This could have indeed been much better explained. To clarify, the full AmazonCat dataset has 13K product category labels, and we assign a reward of one for predicting (recommending) any of the true categories for a given product (context), and zero otherwise. Since we altogether ran a few thousands different configurations of the experiment and had only 1–4 GPUs available to us at any given time, we decided to subset the total number of labels. This forced us to sample the categories based on frequency since otherwise there would be a large number of contexts for which *all* available recommendations would have zero reward (the 100 categories are the same for all the contexts in order to allow nominator specialisation). The choice of 3rd to 101st highest frequency categories then ensures that there are only 5% of those in the training set. The overall proportion of positive rewards in the training set is around 2.25% with the top 5 arms responsible for 19.22% and the bottom 50 for 19.85% of those, a feature of the underlying power law decay (we will include a histogram in the next revision). We provide precision and recall @5 measurements to quantify how well do the models learn the less frequent labels, and observe that MoE outperforms the same model with randomly designed pools.
>
> As for the relation to our theoretical assumptions on data distribution from Section 3, Proposition 1 does not constrain context distribution, and Proposition 2 constructs two specific adversarial context distributions for the case with fixed action pools. The MoE approach from Section 4 then aims to avoid the issues identified in Proposition 2 by learning the pool allocation instead of keeping it fixed, while still allowing each nominator to specialise (which should mitigate the typically lower modelling flexibility of the nominators); hence, as mentioned above, “Section 3 directly motivate[s] the MoE approach in Section 4, but further work is necessary to understand the observed behaviour theoretically.” We will add clarifications in the next revision of the paper.

---

### Author Response · Authors · 2021-08-10
**General response**

We thank the reviewers for their comments and the time they spent assessing our paper. We are glad the reviewers find the topic of our paper highly relevant and underexplored, value our theoretical insights and their integration with the empirical analysis, and appreciate the promise of our Mixture-of-Experts (MoE) based approach to joint training of nominators.

Multiple reviewers asked for further clarification regarding our choice to focus analysis on contextual bandits and simple score-based recommendation algorithms. This decision was made for the following reasons:
* The two-stage setting introduces a number of additional knobs: the number of nominators, the item pool allocation scheme, the architecture & training objective for each nominator and the ranker, and so on. This compounds with the already large number of decisions that need to be made in the single-stage setting, leading to a combinatorial explosion of design choices. Our restriction allows us to put most of the focus on decisions that are unique to two-stage systems.
* Theoretical two-stage recommender literature is sparse to non-existent. Our work is thus a first step on the path to rigorous understanding of two-stage systems. As such, we believe it reasonable and more reader-friendly to start from simpler settings where rigorous discussion is possible without invocation of advanced tools (information-theoretic inequalities, packing numbers, etc.). Furthermore, thorough understanding of simpler cases often provides an excellent jumping-off point for future work that considers more complex algorithms.
* While far from the only option, contextual bandit formulations are reasonably common in the literature [6], including works from Yahoo, Amazon, Netflix, Spotify, and LinkedIn [1,2,3,4,5].

Expanding on the second point, we emphasise that the insights gained in the simplified setting of Section 3 have already been employed to design an improved algorithm in the more complex setting of Section 4 which provides early empirical evidence that pool allocations significantly affect measures other than regret—precision and recall @5—when using an instance of the popular two-tower architecture (the targets are modelled by the dot product between trainable item embeddings, and a trainable affine transformation of the fixed BERT embeddings). The experimental setup in Section 4 is inspired by the “nonlinear factorisation with side information” setting used in the well-cited [7] (and the many follow-up papers).

To facilitate potential discussion, we provide reviewer specific responses separately below.


REFERENCES:

[1] Li, Chu, Langford, Schapire. A contextual-bandit approach to personalized news article recommendation. 2010.

[2] Sawant, Neela, Chitti Babu Namballa, Narayanan Sadagopan, and Houssam Nassif. Contextual multi-armed bandits for causal marketing. 2018.

[3] Kawalle, Jaya, and Elliot Chow. A Multi-Armed Bandit Framework for Recommendations at Netflix. 2018.

[4] Mehrotra, Rishabh, Niannan Xue, and Mounia Lalmas. Bandit based Optimization of Multiple Objectives on a Music Streaming Platform. 2020.

[5] Tang, Liang, Romer Rosales, Ajit Singh, and Deepak Agarwal. Automatic ad format selection via contextual bandits. 2013.

[6] Barraza-Urbina, Glowacka. Introduction to Bandits in Recommender Systems. 2020.

[7] Covington, Adams, Sargin. Deep Neural Networks for YouTube Recommendations. 2016.

---

### Decision · Program_Chairs · 2021-09-27

**Decision:**

Accept (Poster)

**Comment:**

Two-stage recommenders are critically important in production recommender systems, yet as the authors and reviewers point out, there has been very little theoretical analysis of this problem.  While the reviewers do reasonably question whether bandit-based analysis is really the right theoretical framework, the author response rightfully points out that their analysis captures generic characteristics of some deployed two-stage recommenders.  Post-rebuttal, the majority of reviewers agreed with a decision to accept and believe this work can inspire follow-on research in this potentially high impact area.  The authors are strongly encouraged to address review concerns (e.g., on power law and zipf-like distributions) and integrate their insightful rebuttal discussion into the paper (or Appendix), even when it may be self-critical of the present work.